# Association of In-Hospital Mortality and Trauma Team Activation: A 10-Year Study

**DOI:** 10.3390/diagnostics12102334

**Published:** 2022-09-27

**Authors:** Da-Sen Chien, Giou-Teng Yiang, Chi-Yuan Liu, I-Shiang Tzeng, Chun-Yu Chang, Yueh-Tseng Hou, Yu-Long Chen, Po-Chen Lin, Meng-Yu Wu

**Affiliations:** 1Department of Emergency Medicine, Taipei Tzu Chi Hospital, Buddhist Tzu Chi Medical Foundation, New Taipei City 231, Taiwan; 2Department of Emergency Medicine, School of Medicine, Tzu Chi University, Hualien 970, Taiwan; 3Department of Orthopedic Surgery, Taipei Tzu Chi Hospital, Buddhist Tzu Chi Medical Foundation, New Taipei City 231, Taiwan; 4Department of Orthopedics, School of Medicine, Tzu Chi University, Hualien 970, Taiwan; 5Department of Research, Taipei Tzu Chi Hospital, Buddhist Tzu Chi Medical Foundation, New Taipei City 231, Taiwan; 6Department of Anesthesiology, Taipei Tzu Chi Hospital, Buddhist Tzu Chi Medical Foundation, New Taipei City 231, Taiwan; 7Department of Anesthesiology, School of Medicine, Tzu Chi University, Hualien 970, Taiwan

**Keywords:** trauma team, mortality, overtriage, undertriage

## Abstract

Background: Early trauma team activation (TTA) may improve clinical outcomes through early diagnosis and timely intervention by a dedicated multidisciplinary team. Controversy seems to exist about the effect of establishing trauma team systems in traumatic injury populations. Our aim was to identify factors that may be associated with clinical outcomes in trauma injury and to investigate the effect of trauma team activation. Method: This retrospective descriptive study included all traumatic patients from the Taipei Tzu Chi Hospital Trauma Database. All prehospital vital signs, management, injury type, injury mechanisms, hospitalization history, and clinical outcomes were analyzed, and multivariable logistic regression was used to investigate the association between trauma team activation and clinical outcomes. Subgroups of TTA in minor injury and non-TTA in major injury were also analyzed. Result: In this study, a total of 11,946 patients were included, of which 10,831 (90.7%) patients were minor injury (ISS < 16), and 1115 (9.3%) patients were major injury (ISS ≥ 16). In the minor injury population, TTA had a higher intensive care unit (ICU) admission rate, operation rate, re-operation rate, and prolonged total length of stay (LOS). In the major injury population, TTA had a higher mortality rate, prolonged total LOS, and prolonged ICU LOS. After adjusting for mechanism of injury and injury severity, there was no association between in-hospital mortality and TTA, compared with the non-TTA group. However, the TTA group had a higher risk of ICU admission, prolonged ICU LOS, and prolonged total LOS. The subgroup analysis showed trauma team activation had a higher risk of mortality in the 60- to 80-year-old population, major injury (ISS ≥ 16), consciousness clear population, and non-head injury group. Conclusions: We found there was no significant association between in-hospital mortality and TTA. However, in the TTA group, there was a higher risk of ICU admission, prolonged total, LOS, and prolonged ICU LOS. In the subgroup analysis, TTA had a higher risk of mortality in the 60- to 80-year-old population, major injury (ISS ≥ 16), consciousness clear population, and non-head injury group. Our results reflect TTA-criteria-selected patients with greater ISS and a high risk of mortality.

## 1. Introduction

Trauma is the leading cause of death and disability [1]. In the current concept, early diagnosis and timely intervention are promoted in traumatic populations, especially in major traumatic injury populations, which may improve clinical outcomes and prevent catastrophic complications. A dedicated multidisciplinary team is necessary for early diagnosis and timely intervention of traumatic injury patients. A “Trauma Team”, is introduced, consisting of surgeons, emergency physicians, anesthetists, and nurses, led by a team leader to improve outcomes for patients who have suffered severe injuries. It has become standard practice for hospitals in Taiwan. Early trauma team activation (TTA) within 10 min is a quality indicator project for the accreditation of hospitals. This is based on previous studies showing improved mortality among severely injured trauma patients [2,3,4,5,6,7]. In a study by Nirula et al. [8], functional outcomes of minimally penetrating trauma injury and all injured blunt trauma patients receiving a tiered trauma care system had a higher likelihood of total independence. According to the summary by Celso et al. [9], trauma system care could reduce the mortality rate in trauma injury by 15%, as confirmed in large population-based studies. Therefore, delayed or non-activation of the trauma team may increase mortality or decrease functional outcomes [3]. However, data from Connolly et al. [4] showed no significant difference in mortality, median length of stay (LOS), or median time to operative management in TTA delay of more than 30 min. Controversy seems to exist about the effect of establishing trauma team systems in the traumatic injury population. We aimed to identify factors associated with clinical outcomes in trauma injury, investigate the effect of TTA, and assess the factors and analyze the impact of undertriage and overtriage of TTA.

## 2. Methods

### 2.1. Study Setting and Patients Data Source

This is a retrospective cohort study approved by the Institutional Review Board of Taipei Tzu Chi Hospital approval for this study (IRB Number: 10-XD-079). We analyzed all patients with traumatic injuries from the Taipei Tzu Chi Hospital Trauma Database, including patients from January 2009 to 2019 who had visited Taipei Tzu Chi Hospital with hospitalization histories due to traumatic injury. The exclusion criteria were patients without hospitalization or under 20 years old. Detailed demographic, TTA, and clinical outcomes were collected from the trauma database, computerized records, and charts. The prehospital collected demographic data were age and sex, comorbidity, injury location, injury mechanism, prehospital vital signs, and EMT treatment. In terms of mechanism of injury, we defined three major mechanisms for analysis such as road transport, low fall (less than 2 m), high fall (more than 2 m), and others (burn and drowning, etc.). In-hospital parameters included triage, TTA, in-hospital vital signs, and emergent treatment. Injury severity was analyzed by four major scores, including the Injury Severity Score (ISS), the Revised Trauma Score (RTS), the New Trauma and Injury Severity Score (TRISS), and the New Injury Severity Score (NISS). The ISS and RTS were adopted as indices of trauma severity [10,11]. We defined major traumatic injury as ISS ≥ 16 and RTS < 7. We also used the shock index with a cut-off value of 1 to dichotomize shock status [12]. The clinical outcomes were analyzed via hospitalization time, ICU admission, re-admission ICU, ICU admission time, operation, re-operation, and mortality. In clinical outcomes, we defined prolonged total LOS as more than 14 days and prolonged ICU LOS as more than 7 days. Patients with an ISS ≥ 16 without TTA are defined as undertriage, and patients with an ISS < 16 with TTA are defined as overtriage.

### 2.2. Trauma Team Establishment

Trauma care should be a team sport to be accomplished by an organized team, which may make resuscitation more effective. Our trauma team consisted of experienced attending physicians or surgeons, including general surgeons, orthopedic surgery, neurosurgery, and emergency physicians. Participants should be regularly trained and have passed the Advanced Trauma Life Support^®^ (ATLS^®^) program by the American College of Surgeons (ACS) and its Committee on Trauma (COT) [13,14]. Furthermore, participants should undergo trauma continuing medical education every year for at least 8 h. Trauma general ward and trauma ICU nurses should also receive trauma continuing medical education every year for at least 8 h. In addition, TTA cases should be followed up by trauma case managers. An interdepartmental meeting should be held every month to review the trauma team activation cases, undertriage cases, and overtriage cases. Evaluation indicators, including the time from TTA to trauma team leader arrival, the time from admission to surgery in major trauma population, the percentage of trauma team activation, the time from admission to hospitalization, and the percentage of mortality and morbidity, should be reviewed and analyzed every three months.

### 2.3. Trauma Team Activation Criteria

Trauma team activations occurred according to predetermined and institution-specific criteria shown in Table 1. The trauma team activation criteria were modified from guidelines published by the Committee on Trauma of the American College of Surgeons [15]. The criteria were based on three major components, including physiology, anatomy, and mechanism of injury, which reflect the highest-level traumatic injury [16]. Trauma teams could be activated from emergency medical systems, triage staff, and emergency physicians. Prehospital activation by emergency medical systems provides early setting and preparation for resuscitation. In triage, patients with traumatic injury with level I triage are one of the absolute indications for TTA. After emergency physician surveying, physicians could also activate trauma teams based on specific injuries or high-risk injury mechanisms. In our TTA criteria, patients have three chances for early detection of high-risk traumatic injury. After TTA, an experienced trauma physician would arrive within 10 min and facilitate resuscitation, diagnosis, and definitive treatment. The flow chart for TTA is shown in Figure 1.

### 2.4. Statistical Analysis

All continuous data were tested for normal distribution using the Kolmogorov–Smirnov test. All dichotomous and categorical variables are presented as sample numbers with percentages (n, %). Continuous variables are shown as mean with standard deviation (mean ± SD). For comparison of continuous variables, non-parametric ANOVA or Mann–Whitney *U* test was used. Pearson’s chi-squared test or Fisher’s exact test was used to analyze categorical and nominal variables. Multivariable logistic regression was used to investigate the association between parameters and clinical outcomes in the traumatic injury population. Variables with *p* < 0.10 or important factors were selected for multivariable logistic regression analysis. In the subgroup analysis, multivariable logistic regression was used via SPSS software (version 13.0 SPSS Inc., Chicago, IL, USA) for statistical analysis. Statistical significance was defined as *p* < 0.05.

## 3. Results

### 3.1. Characteristics of Study Objects

A total of 11,946 patients with trauma were included, and detailed demographic characteristics are listed in Table 2. In total, 480 (4.0%) patients were TTA with 158 (32.9%) minor injuries and 322 (67.1%) major injuries based on ISS ≥ 16. In the non-TTA group, 10,673 (93.1%) were minor injuries, and 793 (6.9%) were major injuries. In our criteria, the undertriage rate (non-TTA with ISS ≥ 16) was 6.9% (793/11,466 in non-TTA), and the overtriage rate (TTA with ISS < 16) was 32.9% (158/480 in TTA).

In the TTA group, the age was younger, with 48.30 ± 19.21 in patients with minor injuries and 52.57 ± 19.57 in patients with major injuries, than in the non-TTA. The age distribution showed that the TTA group was younger than the non-TTA group (median ± SD: 51.16 ± 19.54 vs. 59.68 ± 20.26, *p* < 0.001). Triage level was higher in the TTA group than in the non-TTA group. The male population was major in the TTA group but not in the non-TTA group. In the TTA group, 229 (47.7%) patients were unconscious at triage, and 66 (18.2%) presented with shock. Severity of injury also showed more severe in the TTA group with a higher proportion of RTS < 7 (317 patients, 66.0% vs. 494 patients, 4.3%; *p* < 0.001), NISS (31.75 ± 25.40 vs. 8.40 ± 6.76; *p* < 0.001), and TRISS (0.62 ± 0.41 vs. 0.96 ± 0.13; *p* < 0.001). In injury mechanism analysis, road transport accounted for 50.4% in the TTA group, followed by high fall with 24.8%. Low fall is more common in the non-TTA group with 40.9% than in the TTA group.

### 3.2. In-Hospital Mortality and Other Clinical Outcomes

In clinical outcome analysis, ICU admission rate [281 (58.5%) vs. 1652 (14.4%); *p* < 0.001], re-admission ICU rate [7 (1.5%) vs. 37 (0.3%); *p* < 0.001], re-operation rate [52 (10.8%) vs. 299 (2.6%); *p* < 0.001], surgical complications rate [77 (16.0%) vs. 1145 (10.0%); *p* < 0.001], and in-hospital mortality rate [163 (34.0%) vs. 200 (1.7%); *p* < 0.001] are all higher in the TTA group than in the non-TTA group. In total hospitalization days and ICU LOS analysis, we found a high proportion of prolonged total LOS (total LOS ≥ 14 days) and prolonged ICU LOS (ICU LOS ≥ 7 days) in the TTA groups. In the minor injury population, there was no significant difference in re-admission ICU, prolonged ICU LOS, and mortality between the non-TTA group and the TTA group. However, the non-TTA group had a high risk of ICU admission. In the major injury population, the TTA group had a higher risk of prolonged ICU LOS and mortality (Table 2).

We performed a quantitative assessment of associations between TTA and clinical outcomes by performing an odds ratio (OR) analysis in Table 3. There was no significant difference in in-hospital mortality in the TTA group. However, the TTA group has a higher risk of ICU admission (adjusted OR: 2.873; 95% CI: 2.072–3.983), prolonged ICU LOS (adjusted OR: 2.064; 95% CI: 1.483–2.872) and prolonged total LOS (adjusted OR: 1.610; 95% CI: 1.234–2.100) (Table 3).

### 3.3. Annual Progression of TTA with Undertriage and Overtriage

Figure 2 shows that major injuries (ISS ≥ 16) increased yearly. The trend of minor injuries is similar to that of total patients with trauma injuries (Figure 2A). The rate of TTA annually increased and is similar in both the TTA with minor and major injuries (Figure 2B). The mortality rate of the TTA group increased but not in the non-TTA group in all trauma injury populations (Figure 2C). In addition, the mortality rate increased annually in the TTA with major injury group but decreased in the non-TTA with major injury group (Figure 2D). The mortality rate trend was similar between the non-TTA group and the TTA group in the minor injury population. In the ICU LOS and total hospitalization analysis (Figure 3), prolonged ICU and total LOS were higher in the TTA group than in the non-TTA group. The TTA group also had a higher shock index in both minor and major injury populations.

### 3.4. Subgroup Analysis in In-Hospital Mortality

The subgroup analysis showed that TTA had a higher risk of mortality in the 60- to 80-year-old population (adjusted OR: 1.955; 95% CI: 1.041–3.672; *p* = 0.037), severe injury with ISS more than 16 (adjusted OR: 1.815, 95% CI: 1.239–2.660; *p* = 0.002), conscious population (adjusted OR: 5.953, 95% CI: 3.269–10.841; *p* < 0.001), and non-head injury group (adjusted OR: 3.927, 95% CI: 1.442–10.699; *p* = 0.007) (Table 4). There was no association between mortality and TTA in minor injuries, unconsciousness, and isolated head injury.

### 3.5. Undertriage and Overtriage in Isolated TBI and Old Age (Age > 65) Populations

In the isolated traumatic brain injury (TBI) population, the undertriage rate was 23.2%, and the overtriage rate was 1.9%. The undertriage population has a higher ICU admission rate, re-admission ICU rate, operation rate, re-operation rate, surgical complications rate, proportion of prolonged total LOS (total LOS ≥ 14 days), proportion of prolonged ICU LOS (ICU LOS ≥ 7 days), in-hospital mortality rate than the normal TTA group (Table 5). In the multivariable logistic regression analysis, undertriage and overtriage of TTA did not show an association with in-hospital mortality, ICU admission, and total LOS > 14 days. Undertriage was only significantly associated with ICU LOS > 7 days compared with the normal TTA group in the isolated TBI population (Table 6).

In the old age population, the undertriage rate was 7.7%, and the overtriage rate was 0.6%. The undertriage population had a higher ICU admission rate and re-admission ICU rate than the normal overtriage TTA group. The surgical complication rate, proportion of prolonged total LOS, and prolonged ICU LOS of the overtriage and undertriage groups were higher than the normal TTA group. The in-hospital mortality rate was significantly higher in the undertriage than in the normal TTA group. In the multivariable logistic regression analysis, undertriage and overtriage of TTA did not show an association with in-hospital mortality, ICU admission, ICU LOS > 7 days, and total LOS > 14 days.

## 4. Discussion

The present study explored the association between TTA and clinical outcomes in the traumatic injury population. We observed that TTA was associated with a higher proportion of ICU admission, prolonged total hospital LOS, and prolonged ICU LOS. However, there was no significant association between in-hospital mortality and TTA. Our subgroup analysis showed that TTA was associated with higher in-hospital mortality in the 60- to 80-year-old population, major injury (ISS ≥ 16), conscious population, and non-head injury group. In the isolated traumatic brain injury population, TTA and non-TTA were not associated with in-hospital mortality.

Although TTA was not associated with mortality, the TTA group received more trauma care, including ICU care. In a study by Azlan et al. [17], TTA improved trauma care, including reducing the door to operation time. Similar results were reported by Wuthisuthimethawee (2017) [18], who reported that TTA criteria could improve acute trauma care and decrease emergency department LOS. There were several reasons for the results. First, the ATLS program was widely promoted for the concept of trauma care, and trauma team members annually participated in trauma continuing medical education. TTA in a well-educated resuscitation team may not be an important factor in clinical outcomes in patients with trauma. Second, early surgical intervention is the gold standard for trauma injury. The effect of TTA is major on condition stabilization. Our trauma center is fully resourced with 24 h attending surgeons who would participate in all major trauma resuscitations within 10 min of major trauma activation. In addition, resuscitation interventions, such as transcatheter arterial embolization (TAE) or surgery procedure, were available at night and on weekends. [19] This may explain why TTA was not associated with mortality in our hospital. In a study by Connolly et al. [4], they found no clear link between delayed TTA and increased mortality. There was no significant difference in mortality, median LOS, or median time to operative management between the early TTA and delayed TTA groups. Ryb et al. [5] showed similar data that there was no association between delayed TTA and LOS as well as mortality. Third, the clinical outcome of severe trauma injury may not be changed by TTA, such as TBI [20,21]. In our result, TTA may have little effect on the TBI population. Finally, undertriage and overtriage may play an important role in clinical outcomes. TTA was associated with high costs and resources, which should be focused on patients who truly need them. A high proportion of undertriage and overtriage may decrease the effect of TTA. Current guidelines (AAST-COT) recommend an acceptable undertriage rate of less than 5% and an overtriage rate of 25–35% [22]. In our criteria, the undertriage rate (non-TTA with ISS ≥ 16) was 6.9% (793/11466 in non-TTA), and the overtriage rate (TTA with ISS < 16) was 32.9% (158/480 in TTA). The results reflected standard trauma activation criteria may not be adequate to identify the at-risk severely injured trauma patient. Although our results showed a high percentage of undertriage patients up to 6.9% (793 patients) compared to the suggestion of field triage guidelines, undertriage patients also received the definite care as the normal activated group (ICU admission: 66.5 v.s 6.78% and surgical intervention: 37.6% vs. 39.1%) [22]. In the undertriage group, the isolated TBI patients and geriatric patients were two major groups accounting for up to 56.4% (447/793 patients) and 46.5% (396/793 patients). Based on current TBI practice guidelines [23,24,25,26], indications for emergency surgery are based on neurologic status and neuroimaging findings, including hematoma volume, thickness, and evidence of mass effect. Even in TBI patients with intracranial hemorrhage whose ISS score is 16, the surgery may not be necessary for emergency performance. Intracranial hemorrhage patients without indication of surgery would receive close monitoring in ICU such as non-traumatic intracranial hemorrhage groups. In addition, emergency neurosurgery was specifically performed and evaluated by the neurosurgeon. Therefore, the role of TTA in these patients became less and caused undertriage in isolated TBI groups. In the geriatric group, the decrease in physiological reserve increases vulnerability to functional impairment after trauma events and delays the physiological response to trauma stress. Therefore, standard trauma activation criteria may not be suitable to apply in geriatric trauma patients.

Prehospital personnel’s discretion to activate the trauma team could shorten intervention and diagnostic time. Accurately identifying the small proportion of severe trauma injuries needing trauma center access is a challenge for emergency medical services (EMS) providers. There was a marked difference in injury severity, clinical course, and outcomes between prehospital TTA and in-hospital TTA [27]. The prehospital activation trauma team would have a high risk of undertriage and overtriage due to on-scene triage and shorter prehospital time. In addition, the injury or hemodynamic status in the trauma population may become unstable during the transport period. Therefore, suitable prehospital personnel discretion to activate the trauma team may have acceptable undertriage and overtriage rates. To control undertriage and overtriage from prehospital TTA, our trauma team leader and emergency physicians participated monthly in the interdepartmental meeting with EMS providers to review the undertriage and overtriage cases. Although there was a lack of prehospital and in-hospital TTA records in our dataset, the undertriage, and overtriage rates were acceptable, even including prehospital personnel discretion to activate the trauma team in our criteria.

Old age (more than 80), unconsciousness, and isolated head injury may impair risk assessment of major injuries. Age is not a standard TTA) criterion recommended by the Committee on Trauma. However, a study by Bardes et al. [28] found 739 (13.6%) TTAs in elderly patients (aged ≥ 70 years), of which up to 541 (73.2%) were activated based on age alone. Although TTA was based on age, they found 49 (9%) patients died, 149 (27.5%) patients were ISS > 15, 65 (12%) patients underwent immediate intervention, 72 (13%) patients had ED intubations, and 306 (56.6%) patients required admission to the ICU. Using standard TTA criteria to select elderly patients with severe trauma injury may not be appropriate, resulting in potentially dangerous undertriage. In our data, TTA in the elderly population, especially in the 60- to 80-year-old, presented with significantly higher mortality risk. In the 60- to 80-year-old population, the decreased functional residual capacity and increased comorbidity may increase the risk of poor clinical outcomes. The antihypertensive, antiarrhythmic, and anticoagulant medications impair the physiological presentation when injured, which may delay or miss standard TTA criteria.

Previous studies in the TBI population have focused on TBI outcomes based on the injury time to the first CT scan and showed varying results. Diaz et al. [29] found no difference in mortality or adverse discharge disposition between TTA and trauma team consultation in traumatic isolated intracranial hemorrhage in elderly patients with anticoagulation use. However, TTA was associated with a more rapid evaluation and diagnosis. In isolated brain injury populations, lower admission GCS was the only factor independently associated with increased risk of death, and any alteration in GCS was strongly associated with mortality. In addition, TBI with TTA has a faster time to CT imaging and anticoagula head injury and unconsciousness were not associated with a high mortality risk between nt reversal; however, there was no significant outcome difference. In our data, isolated TTA and non-TTA. Furthermore, the effect of TT in traumatic isolated head hemorrhage was not larger than in non-isolated hemorrhage.

This study has some strengths. First, our study investigated the effect of TTA in an Asian traumatic injury population, which has not been widely analyzed in previous studies. Second, our study analyzed many important variables and adjusted many essential confounders in the multivariable logistic regression, such as injury mechanism and severity. Third, our study used subgroup analysis to focus on the elderly population, unconscious population, isolated head injury group, and patients with major injuries, the most high-risk population, as a useful guide for prehospital management. Finally, we provided strong evidence that TTA was associated with increased mortality risk, but not in minor injury (ISS < 16), unconscious, and isolated head injury populations. Our major and minor results suggest that TTA in isolated TBI may not be fully activated. We modified the activation in the TBI population by only informing the trauma team leader via cellphone and transferred brain CT. The trauma team could fully activate when the emergency physician or trauma team leader instructs. In elderly patients with trauma, we modified the TTA criteria; if the emergency physician suggests activating the trauma team, the trauma team leader would be informed for activation.

This study has some limitations. First, our study reported several clinical outcomes; however, 30-day mortality and functional outcomes were lacking. Second, there was a lack of records of TTA reasons in this database. Our dataset does not contain information from the time of admission to TTA or trauma team consultation. Third, our retrospective cohort study had inherent issues. Our study was a retrospective study, which only reported the association between important factors and clinical outcomes. Therefore, we could not confirm the results for the causality. However, the large sample size is a strength of our study. Fourth, our trauma database includes mostly blunt injury patients, and the overall penetration group accounted only for 4.4% (528 patients). The TTA in different injury mechanisms (non-penetration vs. penetration) presented different effects on mortality and other clinical outcomes. However, the small sample size of the penetration group caused instability in the subgroup prediction model. Therefore, we did not show the data in our results. Finally, there was a lack of physiological data, which may be useful to reflect risk adjustment of patients with injury to survey undertriage or overtriage.

## 5. Conclusions

This study found that the undertriage and overtriage rates in our criteria were acceptable. There was no significant association between in-hospital mortality and TTA. The TTA group had a higher risk of ICU admission and prolonged total and ICU LOS. In the subgroup analysis, the geriatric and TBI group had a higher undertriage proportion. In terms of undertriage, these patients received similar definite care, such as ICU admission and surgical intervention, as the normal activation groups. Our results showed that the standard TTA-criteria-selected patients had greater ISS and early mortality, but may not be not suitable in the geriatric and isolated TBI population. In addition, the impact on long-term survival may not be appreciated.

## Figures and Tables

**Figure 1 diagnostics-12-02334-f001:**
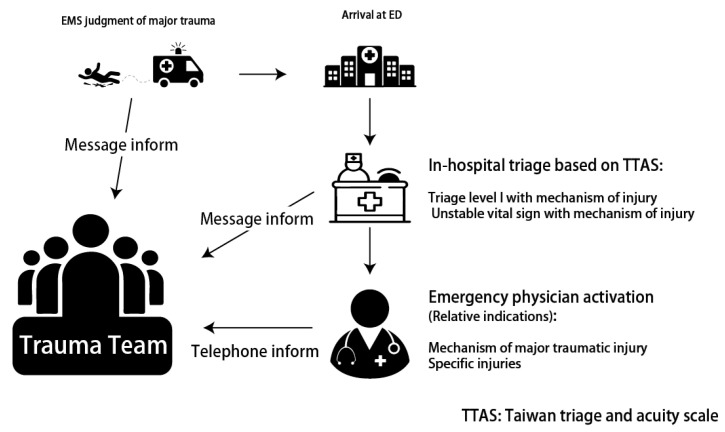
Flow chart of trauma team activation.

**Figure 2 diagnostics-12-02334-f002:**
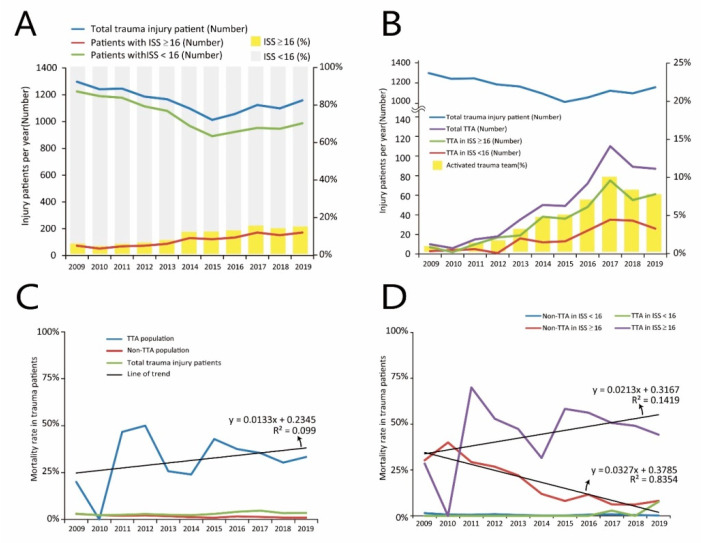
(**A**) Total, minor, and major injury patients number in ten years. (**B**) Trauma team activation with minor and major injury patients number in ten years. (**C**) Annual in-hospital mortality trends in TTA, non-TTA, and total trauma injury population. (**D**) Annual in-hospital mortality trends of TTA and non-TTA in minor and major injury population.

**Figure 3 diagnostics-12-02334-f003:**
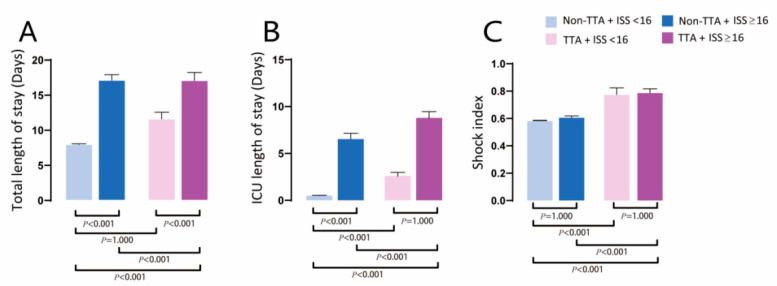
Comparison of (**A**) total LOS, (**B**) ICU LOS, and (**C**) shock index in trauma team activation (TTA) and major trauma (ISS > 16) groups.

**Table 1 diagnostics-12-02334-t001:** Trauma team activation criteria.

Trauma Team Activation Criteria		
Physiological (Absolute Indications):		
Triage level I with mechanism of injury: Cardiac/respiratory arrest Immediate risk to airway: impending arrest Respiratory rate < 10 SBP < 80 (adult) or severely shocked child/infant Unresponsive or responds to pain only (GCS < 9) Ongoing/prolonged seizure Drug overdose and unresponsive or hypoventilation Severe behavioral disorder with immediate threat of dangerous violence		
EMS judgment of major trauma		
Unstable vital signs with mechanism of injury Respiratory impairment Respiratory rate < 10/min or > 29/min Airway obstruction Inability to protect airway Cyanosis or air hunger Paradoxical chest motion Hypotension: SBP < 90 mmHg Altered consciousness or neurological impairment: GCS < 8		
Mechanism of major traumatic injury (Relative Indications):		
Fall from 2 stories or six meters Ejection from vehicle, death in same vehicle High-speed road traffic collisions > 40 km/h Crush injuries torso Penetrating trauma proximal to elbow or knee		
Specific injuries (Relative Indications):		
Severe pelvic fracture with obvious deformity/instability More than two systems injury Altered consciousness or neurological impairment with traumatic injury: GCS < 12 or neurological focal sign		

**Table 2 diagnostics-12-02334-t002:** Comparison of demographic characteristics of patients in the TTA group and the non-TTA group with minor and major injuries.

Characteristics	TTA				Non-TTA				*p*-Value ^‡^
	Total Patients	ISS < 16 (Overtriage)	ISS ≥ 16	*p*-Value	Total Patients	ISS < 16	ISS ≥ 16 (Undertriage)	*p*-Value	
Patient number	480 (4.0%)	158 (32.9%)	322 (67.1%)		11,466 (96.0%)	10,673 (93.1%)	793 (6.9%)		
Age (years)	51.16 ± 19.54	48.30 ± 19.21	52.57 ± 19.57	0.024	59.68 ± 20.26	59.52 ± 20.25	61.84 ± 20.25	0.002	<0.001
Age (years)				0.148				0.004	<0.001
20–40	153 (31.9%)	57 (36.1%)	96 (29.8%)		2223 (19.4%)	2085 (19.5%)	138 (17.4%)		
40–60	156 (32.5%)	56 (35.4%)	100 (31.1%)		3269 (28.5%)	3074 (28.8%)	195 (24.6%)		
60–80	127 (26.5%)	34 (21.5%)	93 (28.9%)		3607 (31.5%)	3340 (31.3%)	267 (33.7%)		
≥80	44 (9.2%)	11 (7.0%)	33 (10.2%)		2367 (20.6%)	2174 (20.4%)	193 (24.3%)		
Sex, n (%)				0.139				<0.001	<0.001
Female	171 (35.6%)	49 (31.0%)	122 (37.9 %)		5528 (48.2%)	5246 (49.2%)	282 (35.6%)		
Male	309 (64.4%)	109 (69.0%)	200 (62.1%)		5938 (51.8%)	5427 (50.8%)	511 (64.4%)		
In-hospital GCS ≤ 8, n	229 (47.7%)	24 (15.2%)	205 (63.7%)	<0.001	196 (1.7%)	59 (0.6%)	137 (17.3%)	<0.001	<0.001
Shock status ^†^, n	66 (18.2%)	22 (14.1%)	44 (21.4%)	0.077	247 (2.2%)	207 (1.9%)	40 (5.4%)	<0.001	<0.001
Triage				<0.001				<0.001	<0.001
1	358 (74.6%)	90 (57.0%)	268 (83.2%)		523 (4.6%)	328 (3.1%)	195 (24.6%)		
2	114 (23.8%)	63 (39.9%)	51 (15.8%)		5982 (52.2%)	5584 (52.3%)	398 (50.2%)		
3	8 (1.7%)	5 (3.2%)	3 (0.9%)		4906 (42.8%)	4708 (44.1%)	198 (25.0%)		
4 and 5	0 (0.0%)	0 (0.0%)	0 (0.0%)		55 (0.5%)	53 (0.5%)	2 (0.3%)		
Injury score systems									
RTS, mean	5.63 ± 2.10	4.98 ± 1.31	4.97 ± 2.10	<0.001	7.74 ± 0.56	7.79 ± 0.37	7.07 ± 1.48	<0.001	<0.001
RTS < 7	317 (66.0%)	68 (43.0%)	249 (77.3%)	<0.001	494 (4.3%)	261 (2.4%)	233 (29.4%)	<0.001	<0.001
ISS	29.32 ± 25.25	7.36 ± 3.73	40.10 ± 24.31	<0.001	7.72 ± 6.18	6.62 ± 2.93	22.51 ± 14.20	<0.001	<0.001
NISS, mean	31.75 ± 25.40	8.56 ± 5.34	43.13 ± 23.54	<0.001	8.40 ± 6.76	7.20 ± 3.37	24.55 ± 15.03	<0.001	<0.001
TRISS, mean	0.62 ± 0.41	0.91 ± 0.21	0.48 ± 0.41	<0.001	0.96 ± 0.13	0.97 ± 0.10	0.83 ± 0.28	<0.001	<0.001
Isolated head injury *	131 (27.3%)	37 (23.4%)	94 (29.2%)	0.182	1802 (15.7%)	1355 (12.7%)	447 (56.4%)	<0.001	<0.001
Injury type				<0.001				<0.001	0.859
Non-penetration	458 (95.4%)	139 (88.0%)	319 (99.1%)		10,960 (95.6%)	10,174 (95.3%)	786 (99.1%)		
Penetration	22 (4.6%)	19 (12.0%)	3 (0.9%)		506 (4.4%)	499 (4.7%)	7 (0.9%)		
Mechanism of injury				<0.001				<0.001	<0.001
Road transport	242 (50.4%)	66 (41.8%)	176 (54.7%)		4121 (35.9%)	3792 (35.5%)	329 (41.5%)		
Low fall	45 (9.4%)	23 (14. %)	22 (6.8%)		4691 (40.9%)	4421 (41.4%)	270 (34.0%)		
High fall	119 (24.8%)	33 (20.9%)	86 (26.7%)		1414 (12.3%)	1279 (12.0%)	135 (17.0%)		
Others	74 (15.4%)	36 (22.8%)	38 (11.8%)		1240 (10.8%)	1181 (11.1%)	59 (7.4%)		
Comorbidity									
CNS diseases	28 (5.8%)	10 (6.3%)	18 (5.6%)	0.745	721 (6.3%)	655 (6.1%)	66 (8.3%)	0.014	0.687
CVD	68 (14.2%)	24 (15.2%)	44 (13.7%)	0.652	3525 (30.7%)	3275 (30.7%)	250 (31.5%)	0.620	<0.001
Respiratory diseases	6 (1.3%)	4 (2.5%)	2 (0.6%)	0.095	273 (2.4%)	253 (2.4%)	20 (2.5%)	0.787	0.108
GI diseases	12 (2.5%)	5 (3.2%)	7 (2.2%)	0.514	316 (2.8%)	299 (2.8%)	17 (2.1%)	0.275	0.737
CKD	5 (1.0%)	3 (1.9%)	2 (0.6%)	0.337	372 (3.2%)	346 (3.2%)	26 (3.3%)	0.955	0.007
Diabetes mellitus	27 (5.6%)	13 (8.2%)	14 (4.3%)	0.083	1499 (13.1%)	1391 (13.0%)	108 (13.6%)	0.637	<0.001
ICU care									
ICU admission	281 (58.5%)	67 (42.4%)	214 (66.5%)	<0.001	1652 (14.4%)	1114 (10.4%)	538 (67.8%)	<0.001	<0.001
Re-admission ICU	7 (1.5%)	0 (0.0%)	7 (2.2%)	0.102	37 (0.3%)	16 (0.1%)	21 (2.6%)	<0.001	<0.001
ICU LOS, days				<0.001				<0.001	<0.001
LOS < 7 days	141 (50.2%)	52 (77.6%)	89 (41.6%)		1226 (74.2%)	912 (81.9%)	314 (58.4%)		
LOS ≥ 7 days	140 (49.8%)	15 (22.4%)	125 (58.4%)		426 (25.8%)	202 (18.1%)	224 (41.6%)		
Surgical intervention									
Operation	182 (37.9%)	61 (38.6%)	121 (37.6%)	0.827	7485 (65.3%)	7175 (67.2%)	310 (39.1%)	<0.001	<0.001
Re-operation	52 (10.8%)	9 (5.7%)	43 (13.4%)	0.011	299 (2.6%)	246 (2.3%)	53 (6.7%)	<0.001	<0.001
Complications	77 (16.0%)	14 (8.9%)	63 (19.6%)	0.003	1145 (10.0%)	954 (8.9%)	191 (24.1%)	<0.001	<0.001
Total LOS				<0.001				<0.001	<0.001
<7 days	212 (44.2%)	65 (41.1%)	147 (45.7%)		6174 (53.8%)	5930 (55.6%)	244 (30.8%)		
7 ≤ days < 14	86 (17.9%)	48 (30.4%)	38 (11.8%)		3852 (33.6%)	3641 (34.1%)	211 (26.6%)		
≥14 days	183 (37.9%)	45 (28.5%)	137 (42.5%)		1440 (12.6%)	1102 (10.3%)	338 (42.6%)		
In-hospital mortality	163 (34.0%)	2 (1.3%)	161 (50.0%)	<0.001	200 (1.7%)	71 (0.7%)	129 (16.3%)	<0.001	<0.001

Definition of abbreviations: CKD: chronic kidney disease; CVD: cardiovascular diseases; ISS: Injury Severity Score; RTS: Revised Trauma Score; NISS: National Industrial Security System; TRISS: New Trauma and Injury Severity Score; LOS: length of stay; and ICU: intensive care unit; ^†^ Shock status: we defined shock condition by shock index more than 1; * Isolated head injury: patients with an AIS code limited to the head and no AIS-coded injury in any other region; **^‡^**
*p*-value: Compared between TTA and non-TTA group.

**Table 3 diagnostics-12-02334-t003:** Multivariable logistic regression of four major clinical outcomes in TTA groups.

Variable	TTA	
	Adjusted OR (95% CI)	*p*-Value
In-hospital mortality	1.218 (0.779–1.904)	0.387
ICU admission	2.873 (2.072–3.983)	<0.001
Prolonged ICU LOS	2.064 (1.483–2.872)	<0.001
Prolonged total LOS	1.610 (1.234–2.100)	<0.001

Co-variables used in the multivariable logistic regression included age, sex, Glasgow coma scale, mechanism of injury, Injury Severity Score, and Revised Trauma Score, except the subgroup variable.

**Table 4 diagnostics-12-02334-t004:** Multivariable logistic regression of in-hospital mortality in subgroup analysis.

Variable		TTA	
		Adjusted OR (95% CI)	*p*-Value
	Age		
	20–40	1.583 (0.721–1.891)	0.252
	40–60	1.595 (0.773–3.293)	0.207
	60–80	1.955 (1.041–3.672)	0.037
	≥80 years	1.650 (0.694–3.922)	0.257
	Sex		
	Female	1.921 (1.039–3.553)	0.037
	Male	1.736 (1.113–2.707)	0.015
	Injury score system		
	ISS ≥ 16	1.815 (1.239–2.660)	0.002
	ISS < 16	0.376 (0.097–1.447)	0.155
	Glasgow coma scale (GCS)		
	GCS < 8	1.038 (0.669–1.611)	0.869
	GCS ≥ 8	5.953 (3.269–10.841)	<0.001
	Isolated head injury	1.233 (0.726–2.096)	0.438
	Non-head injury	3.927 (1.442–10.699)	0.007

Co-variables used in the multivariable logistic regression included age, sex, Glasgow coma scale, trauma team activation, injury mechanism, isolated head injury, injury type, Injury Severity Score, and Revised Trauma Score, except the variable of the subgroup.

**Table 5 diagnostics-12-02334-t005:** Comparison of demographic characteristics of undertriage and overtriage patients in isolated TBI population and old age populations.

Variable		Isolated Traumatic Brain Injury				Old Age (Age > 65)			
		Normal Activation	Overtriage	Undertriage	*p*-Value	Normal Activation	Overtriage	Undertriage	*p*-Value
	Patient number	1449 (74.9%)	37 (1.9%)	447 (23.2%)		4692 (91.6%)	32 (0.6%)	396 (7.7%)	
	ICU care								
	ICU admission	714 (49.3%)	22 (59.5%)	328 (73.4%)	<0.001	658 (14.0%)	15 (46.9%)	281 (71.0%)	<0.001
	Re-admission ICU	10 (0.7%)	0 (0.0%)	14 (3.1%)	<0.001	14 (0.3%)	0 (0.0%)	14 (3.5%)	<0.001
	ICU LOS, days				<0.001				<0.001
	LOS < 7 days	552 (77.3%)	14 (63.6%)	187 (57.0%)		489 (74.3%)	8 (53.3%)	165 (58.7%)	
	LOS ≥ 7 days	162 (22.7%)	8 (36.4%)	141 (43.0%)		169 (25.7%)	7 (46.7%)	116 (41.3%)	
	Surgical intervention								
	Operation	213 (14.7%)	4 (10.8%)	123 (27.5%)	<0.001	3130 (66.7%)	10 (31.3%)	120 (30.3%)	<0.001
	Re-operation	18 (1.2%)	1 (2.7%)	16 (3.6%)	0.005	56 (1.2%)	2 (6.3%)	11 (2.8%)	0.002
	Complication	196 (13.5%)	6 (16.2%)	113 (25.3%)	<0.001	438 (9.3%)	5 (15.6%)	93 (23.5%)	<0.001
	Total LOS				<0.001				<0.001
	<7 days	751 (51.8%)	13 (35.1%)	156 (34.9%)		2155 (45.9%)	13 (40.6%)	129 (32.6%)	
	7 ≤ days < 14	392 (27.1%)	7 (18.9%)	117 (26.2%)		1981 (42.2%)	8 (25.0%)	103 (26.0%)	
	≥14 days	306 (21.1%)	17 (45.9%)	174 (38.9%)		556 (11.8%)	11 (34.4%)	164 (41.4%)	
	In-hospital mortality	92 (6.3%)	1 (2.7%)	81 (18.1%)	<0.001	109 (2.3%)	2 (6.3%)	70 (17.7%)	<0.001

**Table 6 diagnostics-12-02334-t006:** Multivariable logistic regression of four major clinical outcomes in isolated TBI and old age populations.

Variable	Isolated Head Injury			
	In-Hospital Mortality	ICU Admission	ICU LOS > 7 Days	Total LOS > 14 Days
	aOR (95% CI)	aOR (95% CI)	aOR (95% CI)	aOR (95% CI)
Trauma team activation				
Normal activation	Ref	Ref	Ref	Ref
Overtriage	0.220 (0.028–1.762)	0.727 (0.339–1.558)	2.037 (0.786–5.284)	1.685 (0.818–3.473)
Undertriage	0.889 (0.430–1.837)	0.381 (0.111–1.311)	0.308 (0.154–0.616) **	0.633 (0.316–1.267)
	**Old Age (age > 65)**			
Trauma team activation				
Normal activation	Ref	Ref	Ref	Ref
Overtriage	1.376 (0.291–6.507)	1.489 (0.659–3.367)	2.179 (0.730–6.500)	1.197 (0.556–2.575)
Undertriage	0.538 (0.265–1.091)	0.426 (0.145–1.256)	0.580 (0.316–1.065)	1.071 (0.591–1.941)

Co-variables used in the multivariable logistic regression included age, sex, Glasgow coma scale, mechanism of injury, injury type, Isolated head injury, triage, shock status, Injury Severity Score, and Revised Trauma Score. ** *p*-value < 0.05.

## Data Availability

Not applicable.

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
