# Peer review of "Association of In-Hospital Mortality and Trauma Team Activation: A 10-Year Study"

_diagnostics, 2022, doi:10.3390/diagnostics12102334_

Round 1

Reviewer 1 Report

Dear authors, thank you for submitting your paper to our joutnal.

The article is original and the topic is of interest. S you remark, truma is lading cause of death and disability and  muldisciplinary early care of these patients, improve prognosis.

Introduction presents propertly the aim of study , objectives are clear and material and methods are well described. Results are described correctly, and the authors have made a strong effort to present them clearly using tables.

Authors improve the discussion by describing  study strengths and limitations. Finally, conclusions sounds.

I´d like to make some suggestion to improved discussion:

ATLS is a trade mark, so please, reknown it as ATLS®. Please, include AAST-COT Guidelines in References List.

Author Response

Dear Reviewers:

We would like to thank the Reviewers for his/her kind expression of our original work. In addition, all the authors are truly appreciative for the Reviewers’ effort, and thank the Reviewer s’ for his/her critical and highly constructive comments on how to improve this manuscript. We are grateful for the opportunity to revise our manuscript; we have endeavored to address the problems indicated by the Reviewers.

Reviewer #1:

Comments and Suggestions for Authors

Dear authors, thank you for submitting your paper to our joutnal.

The article is original and the topic is of interest. S you remark, truma is lading cause of death and disability and muldisciplinary early care of these patients, improve prognosis.

Introduction presents propertly the aim of study, objectives are clear and material and methods are well described. Results are described correctly, and the authors have made a strong effort to present them clearly using tables.

Authors improve the discussion by describing study strengths and limitations. Finally, conclusions sounds.

I´d like to make some suggestion to improved discussion:

ATLS is a trade mark, so please, reknown it as ATLS®. Please, include AAST-COT Guidelines in References List.

We thank the reviewer for checking our article in detail and indicating the unclear information. We have corrected to ATLS ® and included AAST-COT guideline in our manuscript.

Reviewer 2 Report

The aim of the research performed at Taipei Hospital was to identify factors associated with clinical outcomes in trauma patients, investigate the effect of TTA, and assess the factors and analyze the impact of undertriage and overtriage of TTA.

TTA is an established practice in most Trauma Systems around the world and its benefit is clear specifically in severely injured patients. This is also well known fact that most minor trauma patients will survive with or without trauma team involvement.

Authors described an established Trauma System with activation based on ACS criteria.

Statistical analysis showed that TTA was associated with worth outcome, but not with increase mortality, except for elderly group.

In general, higher ISS associated with increase mortality and utilization of hospital resources. It does not have any influence on TTA. Retrospectively, the over- and undertriage was defined as a TTA in patients accordingly to ISS> 16. Being a retrospective parameter, an ISS has no influence in initial triage and TTA.

Obviously, severely injured patients were almost always treated by trauma team and this group has worth outcome (ICU utilization, LOS, but not mortality). I can’t agree with a concept of TTA association with poorer outcome: injury severity associated with an outcome, not TTA.

The strength of the study is not convincing. In order to assess the trauma quality care, mortality is not an endpoint. For example, TTA was associated with shorter time to CT, but there is no need to activate trauma team for isolated head injury?

This misleading concept cause changes in practice and it is not an acceptable policy in major trauma systems around the world.

The TTA is a concept of focused care and resuscitation in trauma patients. Also patients with head injury should have “appropriate” treatment.

In my opinion, the idea was to assess factors associated with TTA. The mortality is often not related to TTA, but to extent of injury and treatment.

TTA was associated with higher ICU admission – this is great. The decision mode by trauma specialist is always preferable, especially in elderly and head trauma (high undertriage rate for this group based on literature and practice).

Practical recommendation:

1. Change the aim – should be clear and achievable.

2. Conclusions should answer the question. ..” no significant increase risk of in-hospital mortality in TTA group” – Do you expect an increase/decrease in mortality?

3. Authors addressed and analyzed many parameters, but most are not related to TTA criteria or treatments changes.

Author Response

Dear Reviewers:

We would like to thank the Reviewers for his/her kind expression of our original work. In addition, all the authors are truly appreciative for the Reviewers’ effort, and thank the Reviewer s’ for his/her critical and highly constructive comments on how to improve this manuscript. We are grateful for the opportunity to revise our manuscript; we have endeavored to address the problems indicated by the Reviewers.

Reviewer #2:

The aim of the research performed at Taipei Hospital was to identify factors associated with clinical outcomes in trauma patients, investigate the effect of TTA, and assess the factors and analyze the impact of undertriage and overtriage of TTA.

TTA is an established practice in most Trauma Systems around the world and its benefit is clear specifically in severely injured patients. This is also well known fact that most minor trauma patients will survive with or without trauma team involvement.

Authors described an established Trauma System with activation based on ACS criteria.

Statistical analysis showed that TTA was associated with worth outcome, but not with increase mortality, except for elderly group.

In general, higher ISS associated with increase mortality and utilization of hospital resources. It does not have any influence on TTA. Retrospectively, the over- and undertriage was defined as a TTA in patients accordingly to ISS> 16. Being a retrospective parameter, an ISS has no influence in initial triage and TTA.

Obviously, severely injured patients were almost always treated by trauma team and this group has worth outcome (ICU utilization, LOS, but not mortality). I can’t agree with a concept of TTA association with poorer outcome: injury severity associated with an outcome, not TTA.

The strength of the study is not convincing. In order to assess the trauma quality care, mortality is not an endpoint. For example, TTA was associated with shorter time to CT, but there is no need to activate trauma team for isolated head injury?

This misleading concept cause changes in practice and it is not an acceptable policy in major trauma systems around the world.

The TTA is a concept of focused care and resuscitation in trauma patients. Also patients with head injury should have “appropriate” treatment.

We thank the reviewer for checking our article in detail and indicating the unclear information. We appreciate the reviewer’s comments and your concern is well taken. We knew that TTA is a concept of focused care and resuscitation in trauma patients. And patients with head injury also should have “appropriate” treatment. However, “appropriate” treatment for isolated TBI can be provided by emergency physician at ED. The role of “neurosurgeon activation” is more important than TTA. In addition, we believed that injury severity associated with an outcome, not TTA. But, after adjusted injury severity and other confounding factors, the results showed TTA was a predictor for mortality in some population, such as GCS ≥ 8 and Non-head injury groups. It means that the role of TTA in these groups is more effective for resuscitation and lower undertriage events were noted.

In my opinion, the idea was to assess factors associated with TTA. The mortality is often not related to TTA, but to extent of injury and treatment.

TTA was associated with higher ICU admission – this is great. The decision mode by trauma specialist is always preferable, especially in elderly and head trauma (high undertriage rate for this group based on literature and practice).

We thank the reviewer for checking our article in detail and indicating the unclear information. We appreciate the reviewer’s comments. Mortality is not very suitable as an endpoint for elevation of TTA. That was our limitations in this study due to lack other time records for intervention. In future, we would record the time as our outcome to investigate the effect of TTA.

Practical recommendation:

  1. Change the aim – should be clear and achievable.

We thank the reviewer for checking our article in detail and indicating the unclear information. We have added more describe in our article.

  1. Conclusions should answer the question. ..” no significant increase risk of in-hospital mortality in TTA group” – Do you expect an increase/decrease in mortality?

We thank the reviewer for checking our article in detail and indicating the unclear information. We have corrected our conclusion.

  1. Authors addressed and analyzed many parameters, but most are not related to TTA criteria or treatments changes.

We thank the reviewer for checking our article in detail and indicating the unclear information. We have added more describe in our article to mention these parameters.

Reviewer 3 Report

There are a number of grammatical and editorial mistakes throughout the manuscript. Therefore, editing of English language is required for the manuscript to ensure consistency and accuracy in grammar, punctuation, and etc

No association between in-hospital mortality and trauma team activation.p Value:0.387 not <0.005

TTA criteria selected patients with major trauma and have greater ISS and high risk of mortality

There is a higher risk of mortality in the conscious change populations in other paper 

Trauma mortality in Taiwan 30.4/1000000

traffic accident:14.7/1000000(2014)

Table 2 TTA: triage 3 has 8 person,non-TTA:triage 1, 523 patients did not active Trauma team(triage 1:arrest,shock,severe dyspnea,conscious change,GCS<=9,sustain seizure,OHCA)

line 21 trauma tram is trauma team?

Effectiveness of trauma team on medical resourse utlization and quality of care for patients with major trauma.Wang et al.BMC Health Services Research.(2017)17:505-Trauma team estabolished in September 2010

Author Response

Dear Reviewers:

We would like to thank the Reviewers for his/her kind expression of our original work. In addition, all the authors are truly appreciative for the Reviewers’ effort, and thank the Reviewer s’ for his/her critical and highly constructive comments on how to improve this manuscript. We are grateful for the opportunity to revise our manuscript; we have endeavored to address the problems indicated by the Reviewers.

Comments and Suggestions for Authors

Reviewer

There are a number of grammatical and editorial mistakes throughout the manuscript. Therefore, editing of English language is required for the manuscript to ensure consistency and accuracy in grammar, punctuation, and etc

We thank the reviewer for checking our article in detail and indicating the unclear information. We have received English language editing for our article.

No association between in-hospital mortality and trauma team activation. p Value:0.387 not <0.005

We thank the reviewer for checking our article in detail and indicating the unclear information. We have corrected the mistake.

TTA criteria selected patients with major trauma and have greater ISS and high risk of mortality

We thank the reviewer for checking our article in detail and indicating the unclear information. We have added this concept in our article.

There is a higher risk of mortality in the conscious change populations in other paper. 

We thank the reviewer for checking our article in detail and indicating the unclear information. There are several reasons to explain this result. First, conscious change populations can not directly mention the injury sites which may cause longer time to diagnosis. Second, GCS score is a predictor for poor clinical outcome in trauma patients. Therefore, decrease GCS score in conscious change populations also reflected the poor clinical outcomes. Finally, conscious change population may not just impair by trauma injury. In some groups, such as geriatric and alcohol patients, concurrent other medical problem may also cause conscious change and presented with poor clinical outcomes. Therefore, in our study, we conducted this subgroup analysis to confirm this result. However, further prospective studies is necessary to validate our results.

Trauma mortality in Taiwan 30.4/1000000

traffic accident:14.7/1000000 (2014)

Table 2 TTA: triage 3 has 8 person, non-TTA:triage 1, 523 patients did not active Trauma team (triage 1:arrest,shock,severe dyspnea, conscious change, GCS<=9,sustain seizure, OHCA)

We thank the reviewer for checking our article in detail and indicating the unclear information. In Taiwan, the five-level Taiwan Triage and Acuity Scales (TTAS) system was adapted from the Canadian Triage and Acuity Scales (CTAS) and was found to be a reliable triage system that accurately prioritizes the treatment of patients in the emergency department (ED). The TTAS guidelines recommend a time to physician assessment based on the triage acuity level according to a classification of patients in descending order as follows: level I, resuscitation; level II, emergency; level III, urgent; level IV, less urgent; and level V, non-urgent. It has been estimated that nearly one in three patients who experienced major trauma were undertriaged according to an analysis of 36,395 major trauma patients from the Nationwide Emergency Department Sample of the United States in 2010. Good prioritization of the five-level TTAS system is found with significant differences among patients with major trauma (levels I–III). However, inaccurate prioritization in levels II and III of the TTAS system was found when the patients had high risk of high energy trauma injury. The middle group (level III) is most problematic because it includes the greatest number of patients, but this group had a relatively low mortality rate. These findings may result in physicians not paying close attention to these patients to avoid using limited resources that could be reserved for other potentially sicker patients, which that may make the physician less alert and make the limited resources being taken away from other potentially sicker patients. Therefore, to minimalize undertriage and provide acceptable overtriage rate, the triage level of TTAS would be not fully match the severity of injury. In other words, there are undertriage and overtriage in TTAT or TTA criteria.

line 21 trauma tram is trauma team?

We thank the reviewer for checking our article in detail and indicating the unclear information. We have corrected the mistake.

Effectiveness of trauma team on medical resourse utlization and quality of care for patients with major trauma. Wang et al.BMC Health Services Research.(2017)17:505-Trauma team estabolished in September 2010

We thank the reviewer for checking our article in detail and indicating the unclear information. We have cited and added the concept of this research in our article.

Reviewer 4 Report

Dear authors,

thank you for submitting such an interesting manuscript. Its content is very interesting and helpful to everyday trauma management. 

I feel that in your conclusions, you should include that TTA criteria selected patients with greater ISS and early mortality, but impact on long-term survival may not be appreciated as you do not have long term follow up.

As well, I think you should study and mention separately another subgroup comparing penetrating and blunt trauma.

Finally, more focus should be addressed on the fact that a high percentage of undertriaged patients were in the non-TTA group and that standard trauma activation criteria may not be adequate to identify the at-risk severely injured trauma patient. 

I think you have to expand your discussion section addressing these matters as well. 

Author Response

Dear Reviewers:

We would like to thank the Reviewers for his/her kind expression of our original work. In addition, all the authors are truly appreciative for the Reviewers’ effort, and thank the Reviewer s’ for his/her critical and highly constructive comments on how to improve this manuscript. We are grateful for the opportunity to revise our manuscript; we have endeavored to address the problems indicated by the Reviewers.

Reviewer #3:

Comments and Suggestions for Authors

Dear authors,

thank you for submitting such an interesting manuscript. Its content is very interesting and helpful to everyday trauma management.

I feel that in your conclusions, you should include that TTA criteria selected patients with greater ISS and early mortality, but impact on long-term survival may not be appreciated as you do not have long term follow up.

We thank the reviewer for checking our article in detail and indicating the unclear information. We have added “TTA criteria selected patients with greater ISS and early mortality, but impact on long-term survival may not be appreciated.  “in our conclusion.

As well, I think you should study and mention separately another subgroup comparing penetrating and blunt trauma.

We thank the reviewer for checking our article in detail and indicating the unclear information. We appreciate the reviewer’s comments and your concern is well taken. The TTA in different injury mechanism (non-penetration v.s. penetration) presented different effect on mortality and other clinical outcomes. We have tried to conduct subgroup analysis in injury mechanism (non-penetration v.s. penetration). However, in our trauma database, the overall penetration group accounted only 4.4% (528 patients). The prediction model became unstable and decrease of statistical power was noted. Therefore, we only added this part in our limitation section and did not add any conclusion in this issue. In future, we would conduct prospective study focused on penetration injury to investigate the effect of TTA.

“our trauma database includes mostly blunt injury patients and the overall penetration group accounted only 4.4% (528 patients). The TTA in different injury mechanism (non-penetration v.s. penetration) presented different effect on mortality and other clinical outcomes. However, small sample size of penetration group causes unstable in subgroup prediction model. Therefore, we did not show the data in our results.”

Finally, more focus should be addressed on the fact that a high percentage of undertriaged patients were in the non-TTA group and that standard trauma activation criteria may not be adequate to identify the at-risk severely injured trauma patient.

I think you have to expand your discussion section addressing these matters as well.

We thank the reviewer for checking our article in detail and indicating the unclear information. We appreciate the reviewer’s comments and your concern is well taken. Current guidelines (AAST-COT) recommend an acceptable undertriage rate of less than 5% and an overtriage rate of 25%–35% [23]. In our criteria, the undertriage rate (non-TTA with ISS ≥ 16) was 6.9% (793/11466 in non-TTA), and the overtriage rate (TTA with ISS < 16) was 32.9% (158/480 in TTA). The results reflected standard trauma activation criteria may not be adequate to identify the at-risk severely injured trauma patient. Although our results showed high percentage of undertriaged patients up to 6.9% (793 patients) compared to the suggestion of field triage guideline, undertriaged patients also received the definite care as normal activated group (ICU admission: 66.5 vs 6.78% and surgical intervention: 37.6% vs 39.1%) [23]. In undertriaged group, the isolated TBI patients and geriatric patients were two major group accounted up to 56.4 % (447/793 patients) and 46.5% (396/793 patients). Based on current TBI practice guidelines [24-27], indications for emergency surgery are based on neurologic status and neuroimaging findings, including hematoma volume, thickness and evidence of mass effect. Even TBI patients with intracranial hemorrhage who’s ISS score were 16, the surgery may be not necessary for emergency perform. Intracranial hemorrhage patients without indication of surgery would receive closely monitor in ICU as non-traumatic intracranial hemorrhage groups. In addition, emergency neurosurgery was specific performed and evaluated by neurosurgeon. Therefore, the role of TTA in these patients became less and caused undertriage in isolated TBI groups. In geriatric group, the decrease in physiological reserve increases vulnerability to functional impairment after trauma events and delayed the physiological response to trauma stress. Therefore, standard trauma activation criteria may not be suitable to apply in geriatric trauma patients. We have added the describe in our article.

Round 2

Reviewer 2 Report

Thank you for the revisions and comments made in article. 

Author Response

Thank you for the recommendations for your article.

Reviewer 3 Report

No association between in-hospital mortality and trauma team activation. p Value:0.387 not <0.005,no change

The author did not answer my question,Must reanalyzed

This result no compatible to Title